# Effects of Three Exogenous Substances on Heat Tolerance of Peony Seedlings

Jiaxin Guo [1,†], Yu Huang [2,†], Xingyu Yang [1], Wenxuan Bu [1], Jianing Tian [1], Minhuan Zhang [1,*], Kaili Huang [1], Xiaoning Luo [1,*], Ye Ye [3], Wen Xing [1] and Yating Huang [1]

1   College of Landscape Architecture, Central South University of Forestry and Technology, Changsha 410004, China; 20201852@csuft.edu.cn (J.G.); 20221100344@csuft.edu.cn (X.Y.); 20221100345@csuft.edu.cn (W.B.); 20211200352@csuft.edu.cn (J.T.); 20221200382@csuft.edu.cn (K.H.); xingw426@sina.com (W.X.); 20221200380@csuft.edu.cn (Y.H.)
2   College of Art & Design, Nanning University, Nanning 530200, China; huangyu@nnxy.edu.cn
3   College of Humanities and Arts, Hunan International Economics University, Changsha 410006, China; cfstuzmh@163.com
*   Correspondence: t20040180@csuft.edu.cn (M.Z.); luoxiaoning@csuft.edu.cn (X.L.)
†   These authors contributed equally to this work.

**Abstract:** In this study, one-year-old seedlings of *Paeonia ostii* 'Fengdan' were used as materials. The method of spraying exogenous salicylic acid (SA), calcium chloride ($CaCl_2$) and abscisic acid (ABA) was used. The effects of different concentrations of SA, $CaCl_2$ and ABA on the heat tolerance of peony seedlings under high temperature stress were studied. The optimum concentration and mechanism of SA-, $CaCl_2$- and ABA-induced heat tolerance of peony seedlings under high temperature stress were discussed. The results showed that 100 μmol/L SA, 40 mmol/L $CaCl_2$ and 40 mg/L ABA had the best induction effect on the heat tolerance of peony. The effects of three exogenous substances on the heat resistance of peony seedlings were ranked as SA > $CaCl_2$ > ABA. The three exogenous substances mainly alleviated high temperature damage by alleviating the degradation of chlorophyll (Chl), relative electrical conductivity (Rec) content, increasing superoxide dismutase (SOD) activity, soluble proteins (SPs), free proline (Pro) and soluble sugars (SSs) content, and reducing SSs content under high temperature stress. The five indices had the most significant effect on the three exogenous substances in improving the heat resistance of peony seedlings. These could be used as an index to identify the heat resistance.

**Keywords:** peony; exogenous substances; heat tolerance; seedlings

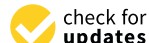



## 1. Introduction

Peony (*Paeonia suffruticosa Andr.*) is a deciduous shrub belonging to Paeoniaceae, Paeonia and *Sect.Moutan*. It is deeply loved by people as a traditional famous flower in China, which has had the reputation of the 'king flower' since ancient times [1]. With global warming, the average temperature in various regions of the world will also increase significantly [2]. High temperature has become one of the adverse environments for plant growth and development. Peony is widely distributed or cultivated in north China, northeast and northwest China because of its preference for cold and dry climate, and its cultivation and application in the Jiangnan area is very limited. High temperature and heat injury have always been an important factor limiting its popularization and application. When the temperature exceeds 40 °C and above, peony cannot withstand the temperature; the growth and development of peony is slow; the plant withers early or even dies; and the disease and insect resistance decreases. The longer the time of high temperature stress, the more serious the heat injury of peony stems and leaves, the more limited the accumulation of organic matter, and the poorer the development of underground roots, which eventually leads to a serious reduction in the ornamental quality of peony [3–5]. The process of

recovering growth after high temperature is slower, and improving the heat resistance of peony is an urgent problem to be solved.

Spraying appropriate concentrations of exogenous substances on different plants has been found to effectively improve the heat resistance of plants [5–8]. A large number of studies have shown that the appropriate concentration of SA, ABA and $Ca^{2+}$ treatment can improve the heat resistance of plants [9]. Dat et al. first reported that exogenous SA treatment could improve the heat resistance of white mustard seedlings [10]. Klein's studies suggest that exogenous $Ca^{2+}$ can promote the synthesis of heat shock proteins and increase the concentration of $Ca^{2+}$ in the cytoplasm during heat shock, thereby improving the heat resistance of crops [11]. Chen Yulong 's research on cotton showed that ABA soaking and root irrigation treatment significantly increased the POD activity of cotton seedlings [12]. SA and $CaCl_2$ treatments could significantly increase the SP content and Pro content of Platycodon grandiflorum under high temperature stress [13]. Yang Huageng et al. showed that calcium and salicylic acid treatment significantly reduced the Rec and MDA content of Phalaenopsis seedlings under high temperature stress [14]. Subsequently, these results were confirmed in many plants.

The materials used in this study are traditional peony varieties, which have good heat resistance and strong knot strength. They can be used as parents for heat-resistant breeding, but there are still different degrees of heat injury in the summer after introduction. Therefore, this study sprayed different concentrations of SA, $CaCl_2$ and ABA under high temperature stress [15]. The chlorophyll content, malondialdehyde (MDA) content, SOD activity, soluble proteins content, free proline content and relative electrical conductivity rate are related to membrane permeability and have been proven to be reliable indicators for identifying heat resistance in different plants. The heat injury index and relative electrical conductivity rate of peony were measured to obtain the optimum concentration of exogenous substances to improve the heat resistance of peony [16,17]. The effects of three exogenous substances (SA, $CaCl_2$, ABA) on the heat resistance of peony seedlings were discussed to obtain the optimum concentration of three exogenous substances to induce the heat resistance of peony seedlings and to establish the mechanism of inducing the heat resistance of peony seedlings. The effect of three exogenous substances on the heat resistance of peony seedlings was obtained, which provided a theoretical reference for solving the heat injury problem of peony.

## 2. Materials and Methods

### 2.1. Study Sites

The outdoor nursery ground is located on the campus of the Central South University of Forestry and Technology in Changsha city in China. The place has a subtropical monsoon climate, characterized by a mild climate, abundant precipitation, rain and heat over the same period, and four distinct seasons. The temperature in the spring changes greatly; there is much rain in early summer; the high temperature in the autumn is long; and the cold in the winter is less severe [18].

The indoor test site is the Garden Plant Laboratory of the School of Landscape Architecture, Central South University of Forestry and Technology. The experimental period is 2020.

### 2.2. Materials

The test plants are the 1-year-old seedlings of the peony variety 'Fengdan' (*P. ostii* 'Fengdan'). The seeds of 'Fengdan' were purchased from the Peony Planting Base of Lijiaping Town, Shaoyang County, Hunan Province, in August 2016. After rooting and then germinating in the sand, the seedlings were planted in a nutrient bowl (1 seedling/pot) of 7 cm × 10 cm × 8 cm (base diameter × top diameter × height). The cultivation substrate was peat: vermiculite: river sand = 3:1:1 (V:V:V), managed by conventional cultivation. After the three leaflets of the compound leaves of the seedlings had

matured, healthy plants with the same height and growth conditions were selected as the biological material for testing.

*2.3. Experimental Methods*

2.3.1. Pretreatment in an Artificial Climate Box

Rinse the test seedlings three times with distilled water and then move them to the neutralizer PQX-450HPL artificial climate box for pretreatment. The temperature in the box was set to 25 °C/20 °C (day/night); the light intensity was 3000 lx; the photoperiod was 12 h/12 h; and the relative humidity (RH) was 80%. After 5 days, the above ground parts of the plants were sprayed with SA solutions at different concentrations: 0.1, 10, 100, 1000, 10,000 μmol/L; $CaCl_2$ solutions at different concentrations: 0.1, 5, 10, 20, 40, 60, 80 mmol/L; ABA solutions at different concentrations: 0.1, 1, 10, 20, 40, 60, 80 mg/L; and sprayed with distilled water as a control. The SA solution, $CaCl_2$ solution, ABA and distilled water used for spraying were adjusted to pH 7 with the NaOH solution. Each plant was sprayed with about 10 mL at 17:00 every day and continuously for 3 days. Then, we placed the seedlings under dark stress at 40 °C for 2 days; the RH was set to 80%; and the seedlings were hydrated and moisturized during the high temperature stress. Nine seedlings were treated with each concentration gradient, and the experiment was repeated three times. After high temperature stress, we measured the relative electrical conductivity of the leaves using Thunderstorm Magnetic DDSJ-308A conductivity meter and observed the heat injury index.

2.3.2. Spraying Three Kinds of Exogenous Substances under High Temperature to Determine Physiological Indices

On the basis of the above experiments, after pretreatment of the peony seedlings used in the experiment, we sprayed distilled water, 100 μmol/L SA solution, 40 mmol/L $CaCl_2$ solution, 40 mg/L ABA solution. The pretreatment and spraying methods were the same as in Test 1. There were 5 test groups in Test 2: blank control (CK): normal temperature (25 °C/20 °C, day/night) + distilled water; HT treatment (HT): high temperature (40 °C/30 °C, day/night) + distilled water; SA treatment (HT + SA): high temperature +100 μmol/LSA; $CaCl_2$ treatment (HT + $CaCl_2$): high temperature + 40 mmol/L $CaCl_2$; ABA treatment (HT + ABA): high temperature +40 mg/L ABA.

We treated 15 seedlings in each test group and repeated the tests 3 times. The light conditions and humidity of the three test groups were the same as in the pretreatment, and the seedlings were hydrated and moisturized every day during high temperature stress. After 6 days of high temperature stress, we moved the plants into a solar greenhouse for recovery. The temperature in the greenhouse was about 25 °C/18 °C (day/night), with natural light and routine management. At day 0, 2, 4, 6 and 7 after high temperature stress recovery, the heat injury index of seedlings was observed with reference to Zhang Jiaping's method [19]. The functional leaves of the same size were taken, and 756 P ultraviolet-visible spectrophotometer (Shanghai Spectral Instrument Co., Ltd., Shanghai, China) was used. The chlorophyll content was determined using the ethanol acetone mixed solution immersion method [20,21]. The MDA content was determined with the TBA chromogenic method [21]. The activity of SOD was determined with the NBT photoreduction method [21]. The content of soluble protein was determined with the Coomassie brilliant blue staining method [21]. The electrolyte permeability of the leaves was measured with the DDSJ-308A conductivity meter [21]. The following formula was used to calculate the index:

$$\text{The heat injury index} = \sum (\text{a certain level} \times \text{the number of plants in this level}) / (\text{the highest level} \times \text{the total number of plants observed in this variety}) \times 100\%. \quad (1)$$

The heat damage classification is as follows: grade 0, the stems and leaves experience almost no heat damage; grade 1, less than 1/4 of the stems and leaves have wilted or burnt;

grade 2, 1/4–1/2 of the stems and leaves have a scorched edge; grade 3, 1/2–3/4 of the stems and leaves have perforated or charred; grade 4, more than 3/4 of the stems and leaves have withered; grade 5, the plant withers or dies.

The smaller the value, the stronger the heat resistance of the plant.

The calculation formula for the chlorophyll content is as follows:

$$C = (12.7 \times D_{663nm} - 2.69 \times D_{645nm}) \times V/(1000 \text{ m}) + (22.9 \times D_{645 \text{ nm}} - 4.68 \times D_{663 \text{ nm}}) \times V/(1000 \text{ m}) \tag{2}$$

In the formula, $D_{663nm}$ and $D_{645nm}$ are the absorbances at the corresponding wavelengths; V is the volume of the extract (mL); m is the leaf mass (g); and the unit of C is mg/g FW.

$$\text{The relative electrical conductivity}(\%) = (EC1/EC2) \times 100\% \tag{3}$$

EC1: Immersion conductivity value;
EC2: Boiled conductivity value.

$$\text{MDA concentrations } (\mu mol/L) = 6.45 \times (A_{450} - A_{600}) - 0.56 \times A_{450}$$

$$\text{MDA content } (nmol/L) = \frac{\text{MDA concentrations } (\mu mol/L) \times \text{extraction volume } (mL)}{\text{sample weight } (g)} \tag{4}$$

$$\text{Free proline content } (mg/g \text{ FW}) = (C \times V/A)/(W \times 1000) \tag{5}$$

C: Proline concentration in the extract ($\mu$g/mL obtained with standard curve);
V: Total volume of extract (mL);
A: The volume of supernatant liquid taken during the determination (mL);
W: Sample weight (g).

$$\text{Soluble sugar content } (mg/g \text{ FW}) = (C \times V/A \times N)/(W \times 1000) \tag{6}$$

C: Sugar content obtained with standard equation ($\mu$g);
A: The sample liquid volume taken during the determination (mL);
V: Total extract amount (mL);
N: Dilution multiple;
W: Sample weight (g).

$$\text{Soluble protein content } (mg/g \text{ FW}) = \frac{C \times V_T}{V_S \times W_F \times 1000} \tag{7}$$

C: The protein content ($\mu$g);
$V_T$: Total volume of extract (mL);
$V_S$: Sample dosage in determination (mL);
W: Sample weight (g).

$$\text{SOD Activity}(U/g) = (A_{CK} - A_E) \times Vt/(W \times V_S \times A_{VK} \times 50\%) \tag{8}$$

$A_{CK}$: Absorbance of light contrast tube;
$A_E$: Sample tube absorbance;
Vt: Total volume of extract (mL);
$V_S$: The amount of sample added during determination (mL);
W: Fresh weight of sample (g).

### 2.3.3. Data Analysis and Processing

Microsoft Office Excel 2013 software was used to sort out the data; Origin Pro9.1 software was used for drawing; and SPSS 20 software was used for variance analysis, correlation analysis and principal component analysis. Referring to the previous methods [22], the effects of three exogenous substances on improving the heat resistance of peony seedlings were comprehensively evaluated with the principal component analysis and membership function method. The following formulae were used.

(1) The calculation of weight

$$D_i = \sum_{i=1, j=1}^{n} (F_{ij} Y_{ij}) \tag{9}$$

$$W_i = D_i / \sum_{i=1}^{n} D_i \tag{10}$$

In the formula, i represents a certain index; $D_i$ represents the effect of each index on improving the heat resistance of peony seedlings by exogenous substances; $F_{ij}$ represents the load of the index on the j th principal component; $Y_{ij}$ represents the contribution rate of the j th principal component; $W_i$ represents the weight of each index.

(2) Calculation of membership function value

The original test data are quantitatively converted by using the membership function formula of fuzzy mathematics. The formula is as follows:

$$f(x)_i = (x_{ij} - x_{imin}) / (x_{imax} - x_{imin}) \tag{11}$$

$$f(x)_i = 1 - (x_{ij} - x_{imin}) / (x_{imax} - x_{imin}) \tag{12}$$

In the formula, $f(x)_i$ represents the membership value of each index of the tested varieties under a certain exogenous substance treatment, and $x_{ij}$, $x_{imin}$, $x_{imax}$ denote the corresponding treatment of each index value, the minimum value of the i index and the maximum value. Many studies have shown that electrolyte permeability is negatively correlated with the heat resistance of plants, that is, the stronger the heat resistance, the lighter the heat damage, the lower the electrolyte permeability, and vice versa. It is a reliable index for evaluating the degree of plant stress. Therefore, this study regards electrolyte permeability as a basic indicator. In the correlation analysis, Formula (11) was used for the negative correlation, and Formula (12) was used for the positive correlation.

(3) Calculation of comprehensive index

According to the membership value of each index and the weight of each index, three kinds of exogenous substances were induced by multiplication.

The comprehensive index (I) of the heat resistance effect of peony seedlings was determined according to the size of the three exogenous substances to induce the heat resistance of peony seedlings.

$$I = \sum_{i=1}^{n} [w_i f(x_i)] \tag{13}$$

## 3. Results

### 3.1. Effects of Exogenous Substances of Different Concentrations on Heat Tolerance of Peony Seedlings

Before the high temperature stress, we sprayed different concentrations of SA. When the SA concentration was 100 μmol/L, the two indicators had the lowest values (Table 1). This shows that 100 μmol/L SA can enhance the heat resistance of peony seedlings and is the most suitable concentration for spraying.

When the $CaCl_2$ concentration applied was 40 mmol/L (Table 2), the minimum values for the determined indicators were obtained, and these were significantly lower compared to the control. This shows that 40 mmol/L $CaCl_2$, as the optimum concentration, can relieve the seedlings of peony under high temperature stress.

**Table 1.** Effects of different concentrations of salicylic acid (SA) on relative electrical conductivity and heat injury index of 'Fengdan' peony seedling leaves.

| SA Concentration (μmol/L) | Relative Electrical Conductivity (%) | Heat Injury Index (%) |
|---|---|---|
| 0 | 84.87 ± 2.14 [b] | 46.67 ± 2.22 [b] |
| 0.1 | 77.69 ± 1.96 [c] | 37.78 ± 4.44 [c] |
| 10 | 69.12 ± 1.36 [d] | 17.78 ± 3.85 [d] |
| 100 | 63.06 ± 2.44 [e] | 5.19 ± 3.40 [e] |
| 1000 | 67.71 ± 3.60 [d] | 17.04 ± 3.40 [d] |
| 10,000 | 96.26 ± 1.64 [a] | 77.04 ± 1.28 [a] |

Note: Values are means ± SD; $n = 225$. Different letters in the same column denote statistically significant differences ($p < 0.05$).

**Table 2.** Effects of different concentrations of calcium chloride ($CaCl_2$) on relative electrical conductivity and heat injury index of 'Fengdan' peony seedling leaves.

| $CaCl_2$ Concentration (mmol/L) | Relative Electrical Conductivity (%) | Heat Injury Index (%) |
|---|---|---|
| 0 | 84.87 ± 2.14 [b] | 44.44 ± 2.23 [b] |
| 0.1 | 78.72 ± 6.97 [ab] | 43.70 ± 1.28 [b] |
| 5 | 73.27 ± 1.03 [bc] | 40.00 ± 2.22 [c] |
| 10 | 69.72 ± 1.92 [cd] | 35.56 ± 2.23 [d] |
| 40 | 63.13 ± 3.94 [d] | 20.74 ± 1.28 f |
| 60 | 83.46 ± 2.66 [a] | 69.63 ± 1.28 [a] |
| 80 | 84.73 ± 4.89 [a] | 66.59 ± 1.86 [a] |

Note: Values are means ± SD; $n = 225$. Different letters in the same column denote statistically significant differences ($p < 0.05$).

A high concentration of ABA (>40 mg/L) will significantly increase the heat injury index of the 'Fengdan'. A low concentration (0.1 mg/L) can significantly reduce the 'Fengdan' heat injury index. This showed that spraying ABA at a concentration of 40 mg/L can improve the heat resistance of peony seedlings (Table 3).

**Table 3.** Effects of different concentrations of abscisic acid (ABA) on relative electrical conductivity (%) and heat injury index of peony seedling ('Fengdan') leaves after high temperature stress.

| ABA Concentration (mg/L) | Relative Electrical Conductivity (%) | Heat Injury Index (%) |
|---|---|---|
| 0 | 84.87 ± 2.14 [a] | 44.44 ± 2.23 [c] |
| 0.1 | 85.11 ± 2.53 [a] | 34.82 ± 1.29 [d] |
| 1 | 64.17 ± 3.64 [b] | 25.93 ± 3.40 [e] |
| 10 | 52.73 ± 3.26 [c] | 26.67 ± 2.23 [e] |
| 20 | 45.27 ± 1.92 [d] | 12.59 ± 2.57 [f] |
| 40 | 40.94 ± 4.13 [d] | 8.15 ± 1.28 [g] |
| 60 | 64.10 ± 5.07 [b] | 58.52 ± 1.28 [b] |
| 80 | 79.87 ± 3.29 [a] | 74.81 ± 3.39 [a] |

Note: Values are means ± SD; $n = 225$. Different letters in the same column denote statistically significant differences ($p < 0.05$).

### 3.2. Comparison of Heat Injury Index of Peony Seedlings under High Temperature Stress

Under high temperature stress, the heat damage degree of peony seedlings increased with the prolongation of stress time. Spraying SA, $CaCl_2$ and ABA could significantly reduce the heat injury index of 'Fengdan' under high temperature stress, and the heat injury index was less than that under the HT treatment. After 7 days of high temperature

stress recovery, the heat injury still showed an upward trend, and the heat injury index decreased after 7 days of application of SA, CaCl$_2$ and ABA (Table 4).

**Table 4.** Comparison of the effects of three exogenous substances on heat injury index of peony seedlings.

| Process | Heat Injury Index (%)/d | | | | |
|---|---|---|---|---|---|
| | 0 | 2 | 4 | 6 | R7 |
| CK | 0 | 0 | 0 | 0 | 0 |
| HT | 0 | 5.37% | 38.24% | 50.43% | 53.75% |
| HT + SA | 0 | 0.49% | 10.00% | 30.67% | 29.00% |
| HT + CaCl$_2$ | 0 | 5.65% | 13.85% | 18.00% | 20.00% |
| HT + ABA | 0 | 6.67% | 26.29% | 40.69% | 43.00% |

Note: R7 means 7 days of growth recovery, the same below. CK represents control check, HT represents high temperature treatment.

*3.3. Comparison of Malondialdehyde (MDA) Content in Leaves of Peony Seedlings under High Temperature Stress*

The spraying of the three exogenous substances all helped in reducing the MAD content of the 'Fengdan', and under high temperature stress, the change trends of MAD all increased first and then decreased. The treatment of all three exogenous substances can help slow down the increase in MDA content in the leaves of peony seedlings under high temperature stress and inhibit membrane lipid peroxidation (Figure 1).

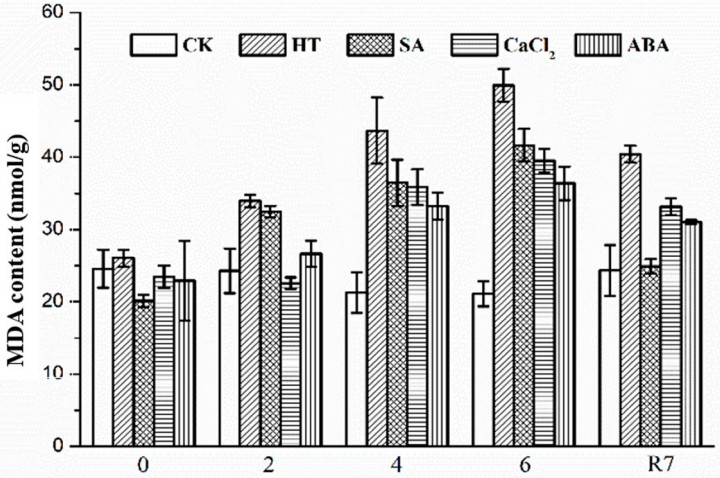

**Figure 1.** Comparison of malondialdehyde (MDA) content in leaves of peony seedlings treated with three exogenous substances. The values are means ± SD; *n* = 225. The x-axis 0, 2, 4, 6 and R7 means the number of spraying days. R7 means 7 days of growth recovery. CK represents control check. HT represents high temperature treatment, the same below.

*3.4. Effect of Optimum Concentration of Exogenous Substances on Relative Electrical Conductivity of Peony Seedling Leaves under High Temperature Stress*

Under the HT treatment, the leaf Rec of peony seedlings increased with the prolongation of high temperature stress, while the spraying of three exogenous substances was significantly lower than that of the HT treatment on the second to sixth days of high temperature stress. Through the spraying of three exogenous substances, on the second day of the treatment, there was no significant difference with CK under the significant difference from the HT treatment; the three exogenous substances treatment was more conducive to reducing the Rec of peony seedlings and slowing down high temperature heat injury. The above results indicate that all three exogenous substances can protect plants under high temperature stress (Figure 2).

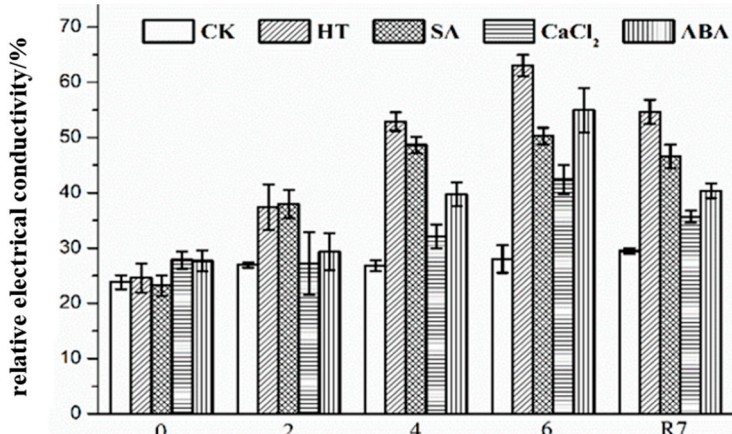

**Figure 2.** Comparison of relative electrical conductivity of peony seedling leaves treated with three exogenous substances. Values are means ± SD; *n* = 225.

### 3.5. Effect of Optimum Concentration of Exogenous Substances on Total Chlorophyll Content (Chl) of Peony Seedling Leaves under High Temperature Stress

Under the SA treatment, the Chl content of 'Fengdan' seedling leaves was significantly higher than that of the HT treatment on the fourth and sixth days of stress. After 7 days of recovery, the seedlings of the 'Fengdan' treated with SA showed a clear recovery trend; the Chl content slightly increased. Under the $CaCl_2$ treatment, the content of Chl in the leaves of 'Fengdan' seedlings increased by 12.99% and 20.08%, respectively, on the fourth and sixth days of high temperature stress. Under the ABA treatment, on the fourth and sixth days of high temperature stress, the Chl content of 'Fengdan' seedlings increased by 12.89% and 20.31% compared with HT. After 7 days of recovery, the Chl content was still significantly higher than that of the HT treatment. The above results indicate that the three kinds of exogenous substances sprayed in the latter period of stress (≥4 d) can effectively alleviate the degradation of Chl in the leaves of peony seedlings under high temperature (Figure 3).

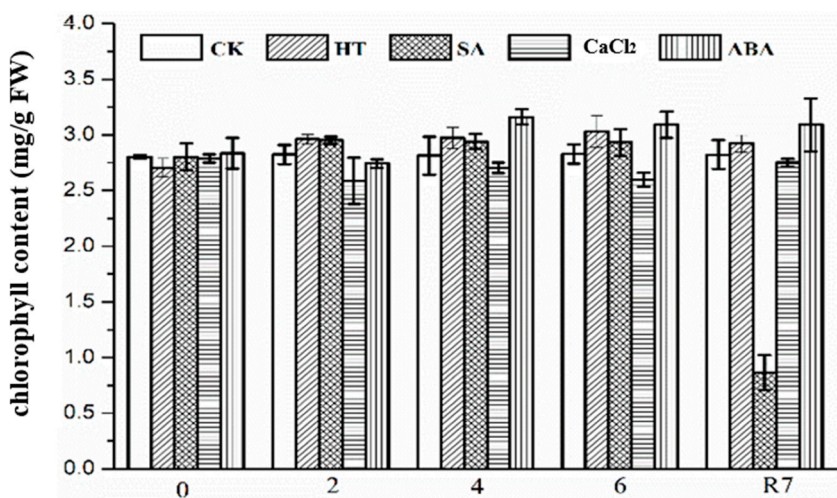

**Figure 3.** Comparison of chlorophyll content in leaves of peony seedlings treated with three exogenous substances. Values are means ± SD; *n* = 225.

### 3.6. Effect of Optimum Concentration of Exogenous Substances on Free Proline (Pro) Content of Peony Seedlings under High Temperature Stress

The spraying of the three substances helped in reducing the content of free proline in the 'Fengdan'. Under the SA treatment, 'Fengdan' seedlings increased by 23.38% and 45.29%, respectively, on the fourth and sixth day of stress compared with the HT treatment.

Under the CaCl$_2$ treatment, the Pro content of 'Fengdan' seedling leaves increased by 27.41%, 8.74% and 85.24% compared with the HT treatment on the fourth, sixth and seventh days of stress. Under the ABA treatment, the Pro content of 'Fengdan' seedling leaves increased by 21.44% and 20.16% on the fourth and sixth day of stress compared with the HT treatment, respectively. The above results indicate that under high temperature treatment, spraying three kinds of exogenous substances can increase the Pro content in the leaves of peony seedlings under stress, especially in the latter period of stress (≥4 d). All three exogenous substances contributed to the restoration of 'Fengdan' in the latter period of stress (Figure 4).

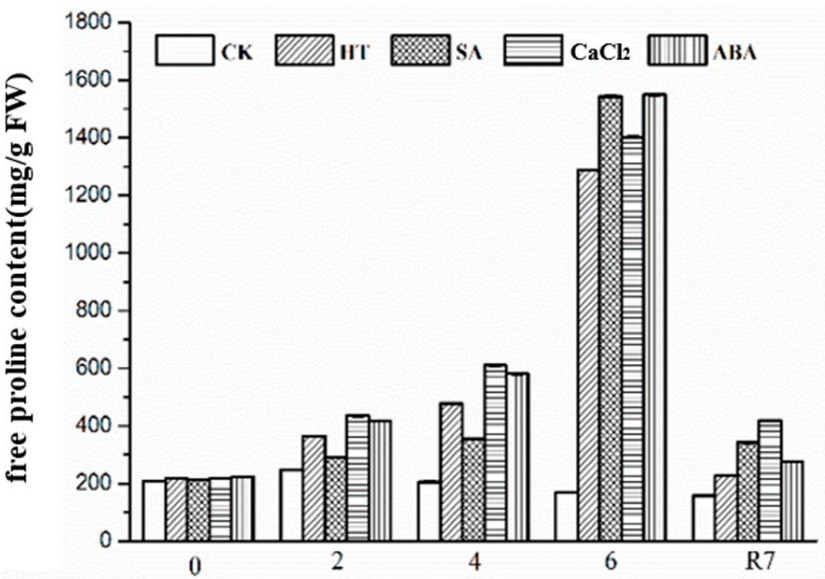

**Figure 4.** Comparison of free proline content in leaves of peony seedlings treated with three exogenous substances.

*3.7. Effect of Optimum Concentration of Exogenous Substances on Soluble Sugar Content of Peony Seedling Leaves under High Temperature Stress*

Under high temperature stress, the change trend of soluble sugars content in peony leaves was exactly the same: it gradually increased with time during high temperature stress and decreased after seven days of resuming cultivation. On the second day of stress, the CaCl$_2$ and SA treatments showed no significant difference. The ABA-treated peony seedlings' soluble sugars (SSs) content was significantly higher than the CK but not significantly higher than the HT treatment. After the fourth day of stress, the content of soluble sugar was significantly reduced under the exogenous substance spray treatment. Under the SA treatment, the seedlings of 'Fengdan' on the fourth and sixth day of stress had SS contents 1.21 and 1.46 times those of the HT treatment. Under the CaCl$_2$ treatment, on the fourth and sixth day of stress, the increase was 44.05% and 29.54% compared with the HT treatment. Under the ABA treatment, the 'Fengdan' seedlings increased by 36.66% and 18.61% compared with the HT treatment on the fourth and sixth days of stress. The above results indicate that spraying all three exogenous substances in the latter period of stress (≥4 d) helped in reducing the content of soluble sugar, thereby reducing the cell penetration potential, and was conducive to alleviating high temperature injury (Figure 5).

*3.8. Effect of Optimum Concentration of Exogenous Substances on Superoxide Dismutase (SOD) Activity of Leaves of Peony Seedlings under High Temperature Stress*

The variation trend of SOD activity as a result of spraying three kinds of exogenous substances was first upward and then downward. After spraying CaCl$_2$ and ABA, the SOD activity was significantly higher than that of the HT group during the high temperature stress treatment; on the fourth day at high temperature, the SOD activity of the leaves

was higher than that of the HT group, but the difference was not significant. After 7 days of recovery, the SOD activity of the 'Fengdan' seedlings sprayed with three exogenous substances rose back to the control level, which was significantly higher than that of the HT group. After the SA treatment, on the second and sixth days of stress, the SOD activity in leaves was significantly higher than that of the HT treatment, increasing by 59.02% and 118.66%, respectively. After treatment with $CaCl_2$, on the second, fourth and sixth days of high temperature stress and seven days after recovery, the SOD activity was 1.88, 1.16, 2.01 and 1.57 times that of the HT treatment, respectively. After the ABA treatment, on the second, fourth and sixth day of heat stress and seven days after recovery, the SOD activity increased by 51.22%, 53.43%, 65.55%, 32.06% compared with the HT treatment. The above results indicate that under high temperature stress, spraying ABA and $CaCl_2$ is beneficial for peony in maintaining higher SOD activity and enhancing antioxidant capacity (Figure 6).

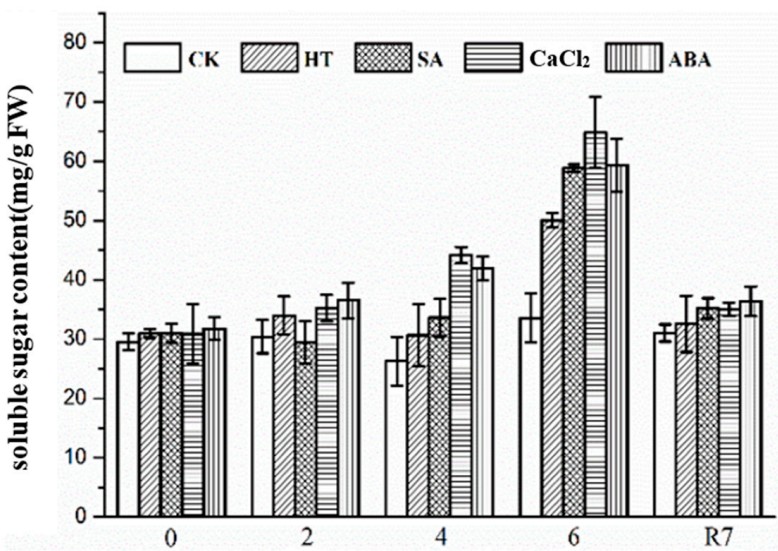

**Figure 5.** Comparison of soluble sugar content in leaves of peony seedlings treated with three exogenous substances. Values are means $\pm$ SD; $n = 225$.

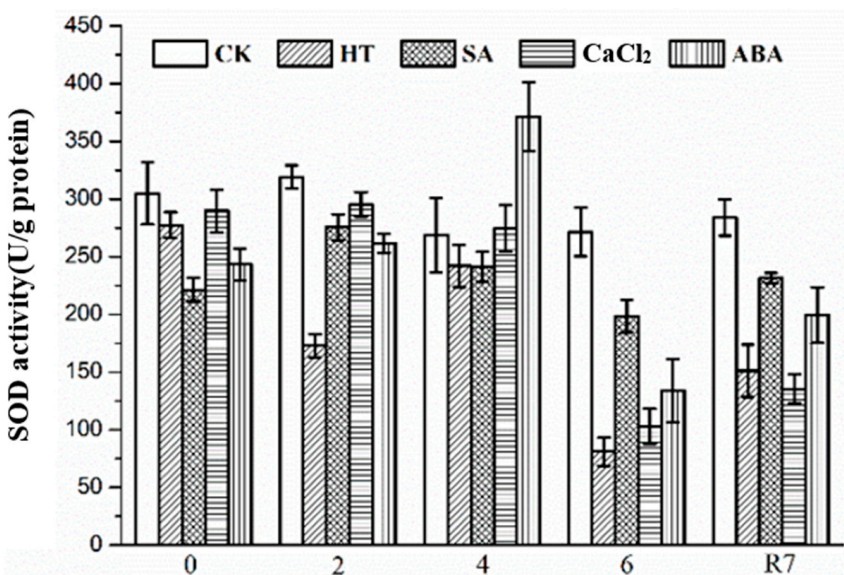

**Figure 6.** Comparison of SOD activity in leaves of peony seedlings treated with three exogenous substances. Values are means $\pm$ SD; $n = 225$.

### 3.9. Effect of Optimum Concentration of Exogenous Substances on Soluble Proteins (SPs) in Leaves of Peony Seedlings under High Temperature Stress

By spraying three exogenous substances in advance, the SPs content of the 'Fengdan' under high temperature stress was significantly higher than that of the HT group. Under the HT treatment and three exogenous substances treatment, the leaf SPs content of leaves increased with increasing stress time. After 7 days of normal cultivation, the SPs contents of seedlings treated with three exogenous substances were significantly higher than the HT group. Under the SA treatment, the SPs content of 'Fengdan' seedlings increased by 34.00%, 17.67% and 19.29% on the second, fourth and sixth days of stress, respectively. Under the CaCl$_2$ treatment, the SPs content of 'Fengdan' at days 4, 6 and 7 of stress was significantly higher than that of the HT treatment, which increased by 15.31%, 20.03% and 41.40%, respectively. Under the ABA treatment, 'Fengdan' increased by 26.36%, 12.66%, 15.23% and 35.89% compared with the HT treatment on the second, fourth, sixth and seventh day of recovery. The above results indicate that pretreatment with the three exogenous substances can increase the SPs content of peony seedlings in the latter period of high temperature stress ($\geq$4 d) (Figure 7).

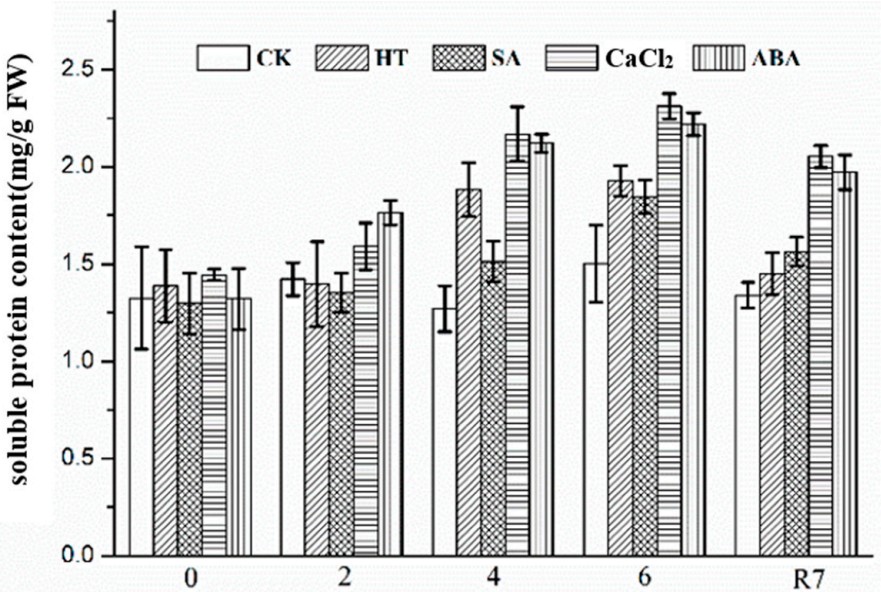

**Figure 7.** Comparison of soluble proteins content in leaves of peony seedlings treated with three exogenous substances. Values are means $\pm$ SD; $n$ = 225.

### 3.10. Comprehensive Evaluation of Heat Tolerance of Peony Seedlings under Optimum Concentration of Exogenous Substances

The SSs content, heat injury index (HII) and Pro content were extremely significantly positively correlated with Rec and significantly correlated with the MDA content; the SOD activity, Chl content and Rec were extremely significantly negatively correlated, while the SPs content was not significantly correlated with Rec. In addition, the SOD activity had a very significant positive correlation with the SPs content; the Pro content had a very significant positive correlation with the SSs content; the SOD activity and SPs content had a very significant negative correlation with the MDA content, etc., and there is a complex correlation between the indices (Table 5).

The five indicators of SOD activity, HII, Chl content, SPs content and Rec have a larger weight value, which has the most significant impact on the three exogenous substances in improving the heat resistance of peony seedlings. Therefore, these can be used as indicators for the identification of heat resistance (Table 6).

**Table 5.** Correlation analysis of eight indices under treatments with three exogenous substances.

| Index | HII | Chl | Rec | MDA | SOD | SPs | Pro | SSs |
|---|---|---|---|---|---|---|---|---|
| HII | 1 | | | | | | | |
| Chl | −0.744 ** | 1 | | | | | | |
| Rec | 0.815 ** | −0.665 ** | 1 | | | | | |
| MDA | −0.018 | 0.350 ** | 0.216* | 1 | | | | |
| SOD | −0.228 * | 0.049 | −0.297 ** | −0.389 ** | 1 | | | |
| SPs | 0.190 * | −0.597 ** | 0.128 | −0.569 ** | 0.268 ** | 1 | | |
| Pro | 0.278 ** | −0.068 | 0.443 ** | 0.270 ** | −0.322 ** | 0.088 | 1 | |
| SS | 0.322 ** | −0.101 | 0.413 ** | 0.189 * | −0.128 | 0.040 | 0.838 ** | 1 |

Note: ** and * mean significant difference at 0.01 and 0.05 levels, respectively.

**Table 6.** Principal component analysis of eight indices under treatments with three exogenous substances.

| Index | Ingredient 1 | Ingredient 2 | Ingredient 3 | F × Y | Weight |
|---|---|---|---|---|---|
| HII | 0.911 | 0.028 | 0.163 | 0.386 | 0.142 |
| Chl | −0.886 | 0.387 | 0.024 | 0.456 | 0.168 |
| Rec | 0.862 | 0.181 | 0.314 | 0.434 | 0.159 |
| MDA | −0.057 | 0.839 | 0.210 | 0.291 | 0.107 |
| SOD | −0.332 | −0.662 | −0.106 | 0.332 | 0.122 |
| SPs | 0.318 | −0.813 | 0.110 | 0.369 | 0.136 |
| Pro | 0.164 | 0.134 | 0.937 | 0.245 | 0.090 |
| SSs | 0.141 | 0.035 | 0.942 | 0.209 | 0.077 |
| Characteristic root | 3.103 | 2.256 | 1.224 | | |
| Contribution rate (%) | 38.791 | 28.204 | 15.300 | | |
| Cumulative contribution rate (%) | 38.791 | 66.995 | 82.295 | | |

Based on the correlation analysis and the principal component analysis, these eight indicators were synthesized, and the membership function value and comprehensive index of each indicator were calculated. The results showed that the best effect of exogenous substances in inducing the heat resistance of peony seedlings was also reached with SA, followed by $CaCl_2$ and thirdly ABA (Table 7).

**Table 7.** Comprehensive index of heat resistance of peony seedlings induced by three exogenous substances.

| Processing | HII | Chl | Rec | MDA | SOD | SPs | Pro | SSs | Composite Index | Sort |
|---|---|---|---|---|---|---|---|---|---|---|
| HT + SA | 0.085 | 0.087 | 0.092 | 0.064 | 0.083 | 0.081 | 0.068 | 0.058 | 0.620 | 1 |
| HT + CaCl$_2$ | 0.088 | 0.080 | 0.097 | 0.068 | 0.071 | 0.081 | 0.064 | 0.056 | 0.606 | 2 |
| HT + ABA | 0.072 | 0.075 | 0.086 | 0.061 | 0.056 | 0.077 | 0.067 | 0.054 | 0.548 | 3 |
| HT | 0.043 | 0.067 | 0.068 | 0.050 | 0.065 | 0.080 | 0.065 | 0.058 | 0.495 | 4 |

## 4. Discussion

### 4.1. Effects of Different Concentrations of Exogenous Substances on Heat Resistance of Peony Seedlings

High temperature stress has adverse effects on plant morphology, physiology and biochemistry during plant growth and development. Under high temperature stress, the growth morphology of plants will undergo a series of changes, such as leaf yellowing, gradual browning, wilting, shedding, and then, whole-plant wilting and even death. The heat injury index is the most direct indicator reflecting the heat resistance of plants. The smaller the value, the stronger the heat resistance of plants, which has been proven in plants such as peony [23] and azalea [24]. In terms of physiology and biochemistry, when the plant is subjected to high temperature stress, the permeability of the plasma membrane increases, the intracellular substances leak, and the relative electrical conductivity increases. Xu Yan showed that Rec was a reliable index for identifying the heat resistance of peony [25]. Zhang Jiaping's study on the heat resistance of peony also showed that Rec had a reliable

reference value due to its small interference factors [19]. Therefore, this paper uses the heat injury index and Rec to screen the optimal concentration of exogenous substances to improve the heat resistance of peony seedlings.

A large number of studies have shown that the optimal concentration of exogenous substances for improving plant heat resistance is related to plant species and seedling age. Wang showed that 100 μmol/L SA could effectively improve the heat resistance of grapes, which was consistent with the results of this study [26]. However, the results of Shen showed that the concentration of 0.05 μmol/L could effectively improve the heat resistance of *Rhododendron hybridum* [24]. Studies have shown that 10 mmol/L $CaCl_2$ can improve the heat resistance of pepper [27], cucumber [28], peanut [29] and other plants. In this paper, the study of peony seedlings showed that 10 mmol/L $CaCl_2$ was not as effective as 40 mmol/L $CaCl_2$ in improving the heat resistance of peony seedlings. Lei Yawei et al. studied the effect of 0~30 mg/L ABA on alleviating the heat injury of three wild *Poa pratensis* germplasms under high temperature stress. The results showed that 20 mg/L ABA treatment had the best effect [30], and this study showed that the best ABA concentration to improve the heat resistance of peony seedlings was 40 mg/L.

*4.2. Induction of Heat Resistance of Peony under High Temperature Stress by Spraying Three Exogenous Substances*

On the one hand, high temperature will block the synthesis of Chl; on the other hand, it will gradually degrade the existing chlorophyll in stems and leaves, and the content will decrease. Zhang Jiaping's study on *Paeonia lactiflora* showed that the Chl content of different *Paeonia lactiflora* varieties decreased with the duration of natural high temperature [19]. Shen reported that the SA treatment could delay the decrease in Chl mass fraction and even increase the mass fraction of chlorophyll [24]. Zai showed that the $Ca^{2+}$ treatment could effectively inhibit the injury of Chl and the decrease in photosynthetic rate of peanut seedlings under high temperature stress [29]. Guo Xiaorui's research on *Catharanthus roseus* showed that exogenous ABA pretreatment could alleviate the injury of heat stress on chlorophyll *a* and *b* of *Catharanthus roseus* seedlings [31]. The results of this study showed that 100 μmol/L SA, 40 mmol/L $CaCl_2$ and 40 mg/L ABA could significantly increase the Chl content in the leaves of *Paeonia ostii* seedlings at the latter stage of stress (≥4 d), which was consistent with the above results. During the whole period of high temperature stress, the Rec of Fengdan treated with three exogenous substances was significantly lower than that of the HT treatment during high temperature stress. The treatment with three exogenous substances could significantly alleviate the increasing trend of MDA content in 'Fengdan' seedlings under high temperature stress. After the seedlings of 'Fengdan' were treated with three exogenous substances, the SOD activity of the leaves increased first and then decreased. Zhang Jiaping also showed similar phenomena in the comparative study of heat resistance of different peony varieties [19]. Under natural high temperature stress, the SOD activity of peony was unstable, and there was no direct correspondence between the heat resistance of peony varieties. Therefore, whether SOD can be used as a reliable indicator of peony heat resistance identification remains to be further studied.

Sun Junli showed that exogenous SA could increase the content of SPs and Pro in grape leaves under high temperature stress [7]. Studies on peanut seedlings showed that the $Ca^{2+}$ treatment relatively increased the content of Pro and SSs in seedlings [29]. Studies have shown that the appropriate concentration of ABA can increase the Pro content and SSs content of wild *Poa pratensis* under high temperature stress [32]; SA and $CaCl_2$ treatments could significantly increase the SPs content and Pro content of *P.grandiflorum* under high temperature stress [13]. This study showed that the SA, $CaCl_2$ and ABA treatments could significantly increase the SPs content, Pro content and SSs content of *Paeonia ostii* at the late stage of stress (≥4 d), which could maintain the low osmotic potential in the cells, increase the concentration of solute in the cells, reduce the water potential and help the cells absorb water continuously from the outside, thus alleviating the high temperature injury.

## 5. Conclusions

The main conclusions of this study were as follows:

1.  After spraying different concentrations of SA, CaCl$_2$, ABA and under stress at 40 °C for 2 days, the heat injury index (HII) and relative electrical conductivity (Rec) decreased first and then increased with the increase in SA, CaCl$_2$ and ABA concentrations. The induction effect of 100 μmol/L SA, 40 mmol/L CaCl$_2$ and 40 mg/L ABA on the heat resistance of peony was the best.

2.  The HII of peony increased continuously with the prolongation of high temperature stress time, and the optimum concentration of SA, CaCl$_2$ and ABA was beneficial to reducing the HII of peony during high temperature stress.

3.  The optimum concentration of SA could significantly increase the stem and leaf dry weight and root dry weight of peony. The chlorophyll content (Chl) decreased continuously with the extension of high temperature stress time. Spraying the optimum concentration of SA, CaCl$_2$ and ABA at the latter stage of stress (≥4 d) could slow down the decline rate of Chl content, reduce Rec and MDA content, increase SOD activity and increase SPs, Pro and SSs content of peony. The Rec and malondialdehyde (MDA) content increased continuously with the prolongation of high temperature stress time; the activity of superoxide dismutase (SOD) decreased first, then increased and then decreased; and the contents of soluble proteins (SPs), free proline (Pro) and soluble sugars (SSs) increased continuously.

4.  SA mainly improved the heat resistance of peony seedlings by alleviating the degradation of Chl under high temperature stress, reducing Rec, reducing MDA content, increasing SOD activity and SPs content. CaCl$_2$ mainly improved the heat resistance of peony seedlings by alleviating the degradation of Chl under high temperature stress, reducing Rec, increasing SOD activity and SPs content. ABA mainly improved the heat resistance of peony seedlings by alleviating the degradation of Chl under high temperature stress, reducing Rec and increasing SPs content.

5.  The significant indices affecting the heat resistance of peony seedlings induced by exogenous substances are HII, Chl content, Rec and SPs content, which can be used as indicators for evaluating the heat resistance of peony seedlings induced by exogenous substances. The effect of three exogenous substances on the heat resistance of peony seedlings was SA > CaCl$_2$ > ABA.

**Author Contributions:** Conceptualization, M.Z.; data curation, X.Y., J.G. and W.B.; experimental design, Y.H. (Yu Huang) and X.L.; review and editing, Y.Y.; experiment instruction, W.X.; resources, J.T., Y.H. (Yating Huang) and K.H. All authors have read and agreed to the published version of the manuscript.

**Funding:** This research was funded by Hunan Natural Science Foundation Project, grant number 2022JJ31010, 2023JJ41035, the sponsors are all Hunan Provincial Science and Technology Department, with a funding amount of 50,000 yuan each, and the Youth Scientific Research Foundation of Central South University of Forestry and Technology, grant number 2019YJ037, the funder is Central South University of Forestry and Technology, with a funding amount of 20,000 yuan, and this research was funded by Hunan Provincial Department of Education Graduate Science and Technology Innovation Project, grant number 2023CX02013, funded by the Education Department of Hunan Province, the amount of funding is 6000 yuan, Hunan Forestry Science and Technology Research and Innovation Fund Project, grant number XLKY202327, the sponsor is Hunan Provincial Forestry Department, and the amount of funding is 200,000 yuan.

**Data Availability Statement:** All datasets supporting the conclusions of this article are included within the article. If not included in the manuscript, they are available from the corresponding author upon reasonable request.

**Acknowledgments:** This work was supported by Hunan Natural Science Foundation Project and the Youth Scientific Research Foundation of Central South University of Forestry and Technology, Institute of Huamn Settlements and Green Low-carbon City and Yuelushan Laboratory Carbon Sinks Forests Variety Innovation Center.

**Conflicts of Interest:** The authors declare no conflict of interest. All authors have read and agreed to the published version of the manuscript.

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
