# Peer review of "Effects of Three Exogenous Substances on Heat Tolerance of Peony Seedlings"

_horticulturae, doi:10.3390/horticulturae9070765_

Round 1
Reviewer 1 Report
General comments
The author’s investigated the effects of the annual seedlings of Paeonia ostii 'Fengdan' and the methods of spraying exogenous salicylic acid (SA), calcium chloride (CaCl2), and abscisic acid (ABA) were used. The effects of different concentrations of SA, CaCl2 and ABA on the heat tolerance of peony seedlings under high temperature stress were discussed. The results show that: 100 μmol / LSA, 40 mmol / L and CaCl2 40 mg / LABA have the best induction effect on the heat tolerance of peony; SA, CaCl2 and ABA mainly improve the heat tolerance of peony seedlings by alleviating the degradation of Chl under high temperature stress, reducing Rec and reducing MDA. The manuscript sounds scientific and hold potential however some points are suggested to improve the overall quality of the manuscript before final publication.
Moderate English editing is required and some typographical errors must be corrected.
Suggestions for authors:
Keywords: Add one or two keywords.
In abstract, rather than general statements, only results should be highlighted to summarize the overall novelty of the manuscript. Rewrite it.
Introduction: The introduction is too short. It must highlight the novelty of your research problem with supportive literature. It should be elaborated.
What standard protocols the authors have followed to prepare different solutions.
Materials and Methods: The biochemical parameters must be explained with their references.
Rectify spacing error throughout the manuscript.
Results:
In figures, what CK, HT, SA stands for? Mention them in caption.
Discussion: It should be more precise and informative. It seems very clumsy. Rewrite the section with latest references.
Conclusion: It should highlight only the major findings of the present study. Rewrite the section.

Author Response
Thank you for your letter and for the reviewers’ comments concerning our manuscript entitled “Effects of Three Exogenous Substances on Heat Tolerance of Peony Seedlings”. Those comments are all valuable and very helpful for revising and improving our paper, as well as the important guiding significance to our researches. We have studied comments carefully and have made correction which we hope meet with approval. Revised portion are marked in yellow in the paper. Our responses to your questions are listed in the attachment.

Reviewer 2 Report
The manuscript ,,Effects of three exogenous substances on heat tolerance of Peony seedlings” is interesting on the first site, but I have a lot of suggestions. Some of them relate to the whole manuscript. The English language needs to be checked for spelling and grammar.
Abstract
I suggest that you shorten the long sentences.
Line 12: Give the Latin name of the species.
Line 14: I was confused about the name of the seedling. What did you use in your experiment, seedlings ‘Fengdan’, ‘Feng Dan’ or ‘Xiangdan’? You only mentioned ‘Fengdan’ here, but later you mentioned two other seedling. Correct this.
Line 14-16: You need to rephrase these sentences.
Line 18: You did not write well… Units of measurement are written together.
100 µmol / LSA is not written well
100 µmol/L SA is written correctly
You have a lot of errors like that, throughout the manuscript. Somewhere you spelled correctly but somewhere you have not. Please correct it.
Line 19: A new sentence starts with SA, CaCl2……..
Line 20: Chl? What is that? Ok, I saw later what it is, but you have to write the name and then the abbreviation. Also Rec and MDA.
Line 23: HII, what is that?
Line 24: The treatment effect…… should be the new sentence.
Line 25: ‘was poor’ delete.
Introduction
Here I have a few suggestions.
1. Somewhere you have two spaces instead of one between words. For example line 32.
2. After full stop you need to put a space. For example line 33.
3. The reference in parentheses is written separately from the text. For example line 33, 36, 37, 39, 40, 41 et cetera.
4. The Introduction section has a lot of repetition. You need to reformulate them.
5. Also, Introduction section is written pretty poor. You need to expand them by information about SA, CaCl2 and ABA; and their effects on seedlings of Peonies and other species.
Materials and Methods
Experimental design is good.
Lines 94 and 95: Heat injury index and electrolyte permeability, how did you calculate them?
Somewhere you mentioned heat injury index (HII) and somewhere heat damage index (HDI) (in table 1). Are they the same or not? If so, you need to use one term.
You did not write at all how you did the analyses: chlorophyll content, electrolyte penetration rate, MDA content, SOD activity, soluble protein content, free proline content, soluble sugar content. You must describe all the methods you used, and put references.
Results and Discussion
Line 117: Correct Dsicussion into Discussion. But, you have separated Discussion section on page 10. Then, you write only Results here.
Line 149: Where is the CK in the Table 4? Also, please rephrase the title of this table.
R7? What is that? Day 7? Then you put only 7 instead of R7.
Line 154: Two peony varieties? In the abstract you mentioned only one.
Line 159: Figure 1….. In this figure, and every another figure, you do not have letters.
Line 190: Figure 3. In the figure, correct CaCl2 into CaCl2. Correct that everywhere.
Line 213: SS, what is this? Put the whole name.
Line 265: ‘Xiangdan’ seedlings?
Lines 279-282: Please put a space between the character ; and another word.
Line 287: Table 6. What are the ingredients 1, 2, and 3?
Line 289-291: Same comment as for Lines 279-282.
Line 303-304: Same comment as for Lines 279-282.
Discussion
In the whole Discussion section you wrote for example Xu Yan et al. and that is ok, but you need to put the year. And also, you need to put that reference in the list at the and of the manuscript.
The English language needs to be checked for spelling and grammar.
Author Response
Dear Reviewer2:
Thank you for your letter and for the reviewers’ comments concerning our manuscript entitled “Effects of Three Exogenous Substances on Heat Tolerance of Peony Seedlings”. Those comments are all valuable and very helpful for revising and improving our paper, as well as the important guiding significance to our researches. We have studied comments carefully and have made correction which we hope meet with approval. Revised portion are marked in yellow in the paper.Our responses to your questions are listed in the attachment.

Reviewer 3 Report
Dear Authors,
Please see the comments below:
Abstract
Kindly italicize species names.
Was the exogenous spraying of the chemicals done separately or was it a mixture?
Please revise the grammar and editing.
Expand on expected contributions to knowledge.
Introduction
Please use scientific statements in sentences.
Lack of references.
Please give a range of the high temperatures in China.
Expand on the physiological effects of heat stress on the Peony.
Why did the study not test the combination of the chemicals?
Materials and method
Give specific descriptions of the climatic conditions eg. Temperature, precipitation\, amount of rain etc.
What sand was used? Does this have any influence on the growth?
Why did you use mature leaves and not younger ones?
Kindly revise the experimental methods, it seems like it is written in instruction, which then is written in person. This section should be written in past tense.
Discussion
Lack of scientific background in this field. Please refer to the relevant literature.
Kindly conduct minor English revisions.
Author Response
Dear Reviewer3:
Thank you for your letter and for the reviewers’ comments concerning our manuscript entitled “Effects of Three Exogenous Substances on Heat Tolerance of Peony Seedlings”. Those comments are all valuable and very helpful for revising and improving our paper, as well as the important guiding significance to our researches. We have studied comments carefully and have made correction which we hope meet with approval. Revised portion are marked in yellow in the paper.Our responses to your questions are listed in the attachment.

Reviewer 4 Report
Dear author,
I kindly ask you to consider my observations and recommendation that I have done on the attached revised manuscript. There are many changes to done. Also, come clarficications are needed. For example, you have studied only one cultivar and some comments of your results imply two ccultivars.
For example:
The study objectives have not clear presented at the end of the Introduction section.
The section Materials and Methods does not offer sufficient information.
Tables and Figures ----- explanations.
Conclusions sholud be concise.
Citation in text and Reference section should be done according to the Instructions for authors.
My goal was to come to your support.
Thank you so much for understanding and collaboration,

Author Response
Dear Reviewer 4:
Thank you for your letter and for the reviewer 4’ comments concerning our manuscript entitled “Effects of Three Exogenous Substances on Heat Tolerance of Peony Seedlings”. Those comments are all valuable and very helpful for revising and improving our paper, as well as the important guiding significance to our researches. We have studied comments carefully and have made correction which we hope meet with approval. Revised portion are marked in yellow in the paper.Our responses to your questions are listed in the attachment.

Round 2
Reviewer 2 Report
Dear authors,
First of all, thank you for your thoughts on improving your manuscript.
I saw that you added three authors, ok. I hope they participated in the experiments.
I have revised the manuscript and I have some suggestions again.
This manuscript has technical errors. Some of them I wrote last time. I suggest that in the future you reed somewhere about how you should write.
Abstract
Line 13: Please delete the spaces between the word and the apostrophe (' Fengdan ').
Line 19 and 20: At the end of the sentence goes full stop, not ; sign.
Line: 22: Chlorophyll index (Chl)? I think this is the chlorophyll content.
Line 27: Seedlings instead of seedings. Correct this.
Introduction
Line 30: You have two spaces again instead of one, correct that.
The Introduction section is better written, but it still lacks the question of how these substances affect other species, a result.
Line 53: I think it is the chlorophyll content.
Materials and Methods
Line 71-80: This part (yellow) is unnecessary. This is weather information and that does not belong here. It's just redundant.
Line 90: Delete spaces (V:V:V).
Line 85-92: I do not understand…. 1 year old seedlings, rooting and germination….. 1 year old seed or seedling?
Line 132: chlorophyll index?
Line 132-137: Ok, you mentioned all the methods you did and gave references, but when I went to see those methods, I did not find those references.
Results
Line 172: Table 4 – I think you should reformulate the title. For example, Heat injury index (%)/day instead of Processing time/d.
Line 185: Where are the letters in the graphs (Figure 1)? I can not see them. Also, figures 2, 3, 4, 5, 6 and 7.
There are still technical errors. I can not repeat them again.
Line 233: Where are the SDs on graph (Figure 4)?
Discussion
The Discussion section is better written.
Line 345: Latin name put italic.
Line 349: Latin name put italic.
Check technical errors in this section.
I suggest that all authors read this manuscript before you resubmitting it.
Author Response
Dear Reviewer2:
Thank you for your letter and for the reviewers’ comments concerning our manuscript entitled “Effects of Three Exogenous Substances on Heat Tolerance of Peony Seedlings”. Those comments are all valuable and very helpful for revising and improving our paper, as well as the important guiding significance to our researches. We have studied comments carefully and have made correction which we hope meet with approval. Revised portion are marked in yellow in the paper.

Reviewer 4 Report
I recommend to carefully review again the article and consider my observations. There are also come changes to be done for correctness and clarity.

Author Response
Dear Reviewer4:
Thank you for your letter and for the reviewers’ comments concerning our manuscript entitled “Effects of Three Exogenous Substances on Heat Tolerance of Peony Seedlings”. Those comments are all valuable and very helpful for revising and improving our paper, as well as the important guiding significance to our researches. We have studied comments carefully and have made correction which we hope meet with approval. Revised portion are marked in yellow in the paper.
